# Provincial Correctional Service Workers: The Prevalence of Mental Disorders

**DOI:** 10.3390/ijerph17072203

**Published:** 2020-03-25

**Authors:** R. Nicholas Carleton, Rosemary Ricciardelli, Tamara Taillieu, Meghan M. Mitchell, Elizabeth Andres, Tracie O. Afifi

**Affiliations:** 1Anxiety and Illness Behaviours Laboratory, Department of Psychology, University of Regina, Regina, SK S4S 0A2, Canada; Nick.Carleton@uregina.ca; 2Memorial University of Newfoundland, Saint John’s, NL A1C 5S7, Canada; eandres@mun.ca; 3Rady Faculty of Health Sciences, University of Manitoba, Winnipeg, MB R3E 0W3, Canada; Tamara.Taillieu@umanitoba.ca (T.T.); Tracie.Afifi@umanitoba.ca (T.O.A.); 4Department of Criminal Justice, University of Central Florida, Orlando, FL 32810, USA; mmitchell@ucf.edu

**Keywords:** mental disorders, Public Safety Personnel, correctional workers, operational stress injuries, posttraumatic stress disorder

## Abstract

Correctional service employees in Ontario, Canada (*n* = 1487) began an online survey available from 2017 to 2018 designed to assess the prevalence and correlates of mental health challenges. Participants who provided data for the current study (*n* = 1032) included provincial staff working in institutional wellness (e.g., nurses) (*n* = 71), training (e.g., program officers) (*n* = 26), governance (e.g., superintendents) (*n* = 82), correctional officers (*n* = 553), administration (e.g., record keeping) (*n* = 25), and probation officers (*n* = 144, parole officers). Correctional officers, workers in institutional administration and governance positions, and probation officers reported elevated risk for mental disorders, most notably posttraumatic stress disorder (PTSD) and major depressive disorder. Among institutional correctional staff, 61.0% of governance employees, 59.0% of correctional officers, 43.7% of wellness staff, 50.0% of training staff, and 52.0% of administrative staff screened positive for one or more mental disorders. In addition, 63.2% of probation officers screened positive for one or more mental disorders. Women working as correctional officers were more likely to screen positive than men (*p* < 0.05). Across all correctional occupational categories positive screens for each disorder were: 30.7% for PTSD, 37.0% for major depressive disorder, 30.5% for generalized anxiety disorder, and 58.2% for one or more mental disorders. Participants between ages 40 and 49 years, working in institutional governance, as an institutional correctional officer, or as a probational officer, separated or divorced, were all factors associated (*p* < 0.05) with screening positive for one or more mental disorders. The prevalence of mental health challenges for provincial correctional workers appears to be higher than federal correctional workers in Canada and further supports the need for evidence-based mental health solutions.

## 1. Introduction

A systematic review demonstrated the limited research on correctional worker’s mental health (i.e., six published studies were identified) [1]. Samples sizes ranged from *n* = 65 to *n* = 3599, with substantial diversity in measurement tools. The results varied substantially (i.e., posttraumatic stress disorder (PTSD; e.g., symptoms of intrusions, avoidance, negative mood, alterations in arousal, and reactivity) 15.0% to 29.1%; major depressive disorder (e.g., depressed mood or diminished interest in activities) 24.0% to 59.7%; and anxiety (e.g., excessive anxiety and worry) 12.2% to 37.9%), including the only Canadian study [2], which was based primarily on federal correctional worker (i.e., Correctional Services Canada; CSC) data rather than provincial workers, and 54.6% screened positive for one or more mental disorders [3].

A subsequent re-analysis indicated no differences in positive screens between occupational categories (i.e., operational institutional; operational community; administrative institutional; administrative headquarters) [4]; however, discrepancies identified in the systematic review [1] may have been based on sample compositions. Institutional correctional workers (e.g., employed in prisons) may directly experience more potentially psychologically traumatic events (PPTEs) than administrators. Probation and parole officers may be directly *and* vicariously exposed to PPTEs. Correctional workers in all areas of correctional services do experience, witness, and come to know details about PPTEs experienced by colleagues, civilians, clients, and custodial populations (e.g., death by suicide, attacks on staff, attacks between prisoners [5,6]), but there may be important differences for correctional workers in provincial employment (e.g., who work with remanded prisoners as well as prisoners who are just off the street) and across occupational categories within any group of correctional workers (e.g., persons managing and involved in diverse elements of prisoner/probationer oversight with different workloads and expectations). In Canada, the federal system is operated by the CSC and responsible for the care, custody, and control of individuals serving sentences of two or more years. Persons remanded into custody (e.g., awaiting trial or sentencing) or serving durations of two years less one day serve in a provincial or territorial correctional facility. Each province and territory has a unique system governed by the provincial or territorial Ministry or Department of Correctional Services [7].

We designed the current study to: (1) provide an initial exploration of self-reported positive screening rates for mental disorders among provincial correctional workers in Ontario, Canada; (2) assess differences in positive screening rates for mental disorders across correctional occupational categories; and (3) explore relationships between positive screening rates for mental disorders and several putative demographic risk factors [2,4], including sex, age, marital status, ethnicity, education, and years of service. 

## 2. Materials and Methods 

### 2.1. Procedure and Data

The current study included data collected from Ontario’ Ministry of the Solicitor General employees. The service employs approximately 8000 people working in 26 provincial correctional institutions (e.g., treatment centers, jails, correctional centers, and detention centers), 100 probation offices and sub-offices, 19 court and institutional offices, and 164 reporting centers [8,9]. Potential participants were emailed by Ministry personnel using two organizational listservs: (1) the Ministry of the Solicitor General; and (2) the Ontario Public Service Employees Union. The emails informed recipients about the survey and invited their voluntary participation. The emails could be forwarded, which means there was no way to definitively identify a sampling frame. An anonymous link was provided in the email that routed the participant to the start of the survey, wherein more detailed information was provided, including informed consent, data confidentiality, data storage procedures, potential risks and resources, and study withdrawal processes. The email could be forwarded, and the email lists likely had some unknown level of overlap; accordingly, there was no way to accurately estimate the number of unique persons successfully invited for potential participation. 

Data were collected using a web-based self-report survey delivered through Qualtrics and available in English or French from 8 December 2017 to 30 June 2018. The survey included well-established screening measures for several mental disorders. The selected measures have been used in previous research examining mental disorders among Public Safety Personnel; (PSP; e.g., communications officials, correctional workers, firefighters, paramedics, police) [2] (Carleton et al., 2018), supporting compatibility with prior PSP research results. Prior to commencement, the study was reviewed for ethical compliance by the Queen’s University and Affiliated Health Sciences Centre Research Ethics Board (file #6024787), as well as the Research Ethics Boards at both the University of Regina (file #2017-098) and Memorial University of Newfoundland (file #20201330-EX). We complied with Canadian Psychological Association ethical standards in the treatment of our sample. We directed all interested persons to a website with study details and were required to explicitly indicate consent before proceeding.

A total of 1487 people began the survey, but only 1032 respondents completed at least some of the mental disorders section. The survey required approximately 25–40 min to complete and could be completed in sections (i.e., respondents could leave the survey and return to complete later). The survey logic allowed participants to skip sections based on responses to screening questions. For example, if a participant responded “no” to the question, “Have you ever been diagnosed with a mental illness”, the participant skipped the associated follow-up questions. Respondents could terminate participation at any time by closing their browser. Respondents provided informed consent by clicking a button that led to the first survey question. 

### 2.2. Self-Report Symptom Measures

Clinically significant mental disorder symptom severity was assessed using the following self-report screening measures: the PTSD Check List 5 (PCL-5; e.g., “Repeated, disturbing, and unwanted memories of the stressful experience?”) [10,11,12,13,14]; the nine-item Patient Health Questionnaire (PHQ-9; e.g., “Feeling down, depressed, or hopeless.”) [15,16,17,18]; the Panic Disorder Symptoms Severity scale, Self-Report (PDSS-SR; e.g., “How many panic and limited symptoms attacks did you have during the past week?”) [19,20,21]; the seven-item Generalized Anxiety Disorder scale (GAD-7; e.g., “Feeling nervous, anxious or on edge”) [18,22,23]; and the Alcohol Use Disorders Identification Test (AUDIT; e.g., “How often do you have a drink containing alcohol”) [24,25]. Participants reported symptoms per the reporting period for each scale which was in the past month for PCL-5; past 14 days for the PHQ-9; past seven days for the PDSS-SR; past 14 days for the GAD-7; and past year for the AUDIT. For the PCL-5, and in line with the Diagnostic and Statistical Manual of Mental Disorders, 5th ed. (DSM-5) [26], participants also reported on their lifetime exposure to a specific list of PPTE provided by the Life Events Checklist-5 (LEC-5) [10,11,12,13,14]. The LEC-5 does not include “Sudden and unexpected death of someone close to you”. Consistent with previous PSP research, some LEC-5 questions were revised; specifically, “Natural disaster” was revised to, “A life-threatening natural disaster”, and a “Transportation accident” was revised to “A serious transportation accident”. Participants were asked to provide details if they selected, “Any other very stressful event or experience”. Participants who endorsed experiencing one or more LEC-5 events were then asked to select a single index trauma (i.e., the single worst traumatic event, most distressing event, or event that was currently causing the most distress) against which they would rate their past month of symptoms using the PCL-5 items. 

A positive screen on the PCL-5 required participants to meet minimum criteria for each PTSD symptom cluster and exceed the minimum clinical cut-off total score of >32 [10]. A positive screen for the other measures was determined based on published recommendations; specifically, a positive screen required the PHQ-9 total score to be >9 [27], the PDSS-SR total score to be >7 [19], the GAD-7 total score to be >9 [28], and the AUDIT total score to be >15 [25]. All measures are validated for screening to identify patients who may require further clinical attention, rather than definitive diagnostic tools.

### 2.3. Self-Reported Diagnostic Status

In addition to the above screening tools, participants were asked whether they had ever been diagnosed with any mental disorder and, if so, asked to report the specific diagnosis. Self-reported diagnoses of persistent depressive disorder, bipolar I, bipolar II, cyclothymic disorder, and “any other mood disorder” were used with the self-report symptom measures to identify the presence or absence of “any mood disorder”. Self-reported diagnoses of social anxiety disorder, obsessive compulsive disorder, and “any other anxiety disorder” were used with the self-report symptom measures to identify the presence or absence of “any anxiety disorder”.

### 2.4. Institutional Categories

Participants were classified into six mutually exclusive occupational categories based on their current position at the time of data collection: (1) Institutional Wellness (*n* = 71; e.g., nurses, social workers, counsellors); (2) institutional training (*n* = 26; e.g., teachers, program officers, chaplains, and coordinator of volunteers, which are all person involved in the coordination and delivery of programming); (3) institutional governance (*n* = 82; e.g., superintendents, deputy superintendents); (4) institutional correctional officers (*n* = 553; e.g., correctional officers in men’s or women’s institutions); (5) probation officers (*n* = 144; i.e., probation and parole officers); and (6) institutional administration (*n* = 25; e.g., administrative assistants and support). 

### 2.5. Socio-Demographic Covariates 

Demographic characteristics included self-reported sex, age, marital status, provincial region, ethnicity, education, years of service, and urban/rural work location. 

### 2.6. Statistical Analyses

Analyses were conducted using SPSS Version 25 software (IBM Corp, Armonk, NY, USA, 2017). Complete case analyses were used throughout. Statistical tests were considered significant at a significance criterion of *p* < 0.05. Descriptive statistics (e.g., means, standard deviations) were calculated for each variable of interest for each occupational group. Logistic regression models were used to assess associations between socio-demographic covariates and any mental disorder among the occupational categories. Post-hoc analyses were computed to compare positive screening frequencies across occupational categories. Post-hoc regression analyses were also computed to assess associations between sex and any mental disorder for each occupational group because of previous research indicating sex differences among some PSP occupations [2].

## 3. Results

Mean scores for all mental disorders across all participants and for each correctional occupational group are in Table 1. Covariates and associations with positive screens are in Table 2. 

Participants were more likely to screen positive for a mental disorder as their age increased until the 50–59 age group, where the proportion decreased, but the only statistically significant difference was between ages 40–49 and ages 20–29 (OR = 1.53, 95% CI = 1.02, 2.28). Participants with four or more years of service were also more likely to screen positive for a mental disorder relative to participants with less than four years of service, but the only statistically significant difference was between those with less than four years of service and those with more than 15 years of service (OR = 0.55, 95% CI = 0.40, 0.76). Participants who reported being single or married/common-law were less likely to screen positive for a mental disorder than participants who were separated/divorced/widowed, but the difference was only statistically significant for married/common-law persons (OR = 2.30, 95% CI = 1.50, 3.54). Participants who completed a university degree or a four-year college program or more education were less likely to screen positive than those who completed high school or less education (OR = 0.78, 95% CI = 0.63, 0.97) (Table 2). 

Participants working in Institutional Wellness were less likely to screen positive than all other categories (Table 2), but the differences were only statistically significant for Institutional Governance (OR, 2.02; 95% CI, 1.06–3.85), Institutional Correctional Officers (OR, 1.85; 95% CI, 1.13–3.05), and Probational Officers (OR, 2.22; 95% CI, 1.24–3.95). Positive screening frequencies for mental disorders based on self-report measures or self-reported diagnostic status for each occupation grouping are in Table 3. Participants screened positive most frequently for PTSD (26%). Several statistically significant differences were identified across correctional occupational categories, as indicated by superscripts in Table 3. There were statistically significant differences between occupational groups, but in general Institutional Wellness personnel were less likely to screen positive than other groups and Institutional Correctional Officers were more likely to screen positive than other groups (Table 3). There were no statistically significant differences in positive screenings across participants working in Institutional Governance, as Institutional Correctional Officers, with Institutional Administration, or as Probational Officers for most disorders (Table 3).

Finally, women were not statistically significant more likely than men to screen positive for a mental disorder overall (Table 2; OR = 1.11, 95% CI = 0.85, 1.45); however, post-hoc analyses comparing the correctional occupational categories (Table 4) indicated that women working as institutional correctional officers were statistically significantly more likely than men to screen positive for a mental disorder (OR = 1.48, 95% CI = 1.04, 2.12).

## 4. Discussion

The current study assessed mental disorder prevalence among Ontario provincial correctional workers [1,2,4] using well-established self-report measures as screening tools. The results are novel, extending previous research with provincial correctional data and nuanced occupational differences [2,4,5,6]. The results can be compared to parallel research with federal correctional workers [2,4].

Across the provincial correctional worker occupational categories, the frequency of positive screens for any mental disorder (i.e., 58.2%) was consistent with results from federal workers (i.e., 54.6%) [2,4], but with higher frequencies for major depressive disorder (i.e., 37.0% versus 31.1%) and generalized anxiety disorder (i.e., 30.5% versus 23.6%). The differences may have been due to variations such as working in remand facilities, overcrowding, or causal or fixed-term employment status [7,29]. There were also differences among provincial participants across occupational categories. Positive screenings for any mental disorder were lowest for persons working in wellness (i.e., 43.7%) and highest for probation officers (i.e., 63.2%). The contrasting patterns for positive screenings may result from diverse provincial correctional occupational roles. For example, persons working in wellness may be more regularly reminded about the importance of adaptative self-care and accessing mental health supports. All correctional workers are frequently exposed to PPTEs [30], but those working inside institutions (e.g., correctional officers, healthcare staff) may perceive themselves as having more accountability than those with less direct contact (e.g., administrators, record keepers, programming officers). Correctional workers with more direct contact may also perceive themselves as having less control, more uncertainty, and more unpredictability, which have been associated with increased mental health risks [31] and may be pronounced for the leadership members (e.g., wardens, superintendents, management) who are ultimately responsible. In contrast, community correctional workers (e.g., probation officers) may be exposed both directly (e.g., witnessing) and indirectly (e.g., reading about) to PPTEs, with variable levels of perceived control.

There were several sociodemographic factors associated with positive mental disorder screenings. Women correctional officers were more likely than men to screen positive for any current mental disorder, but there were no such differences for other occupational categories. As in other PSP professions [32,33,34,35,36], systemic variables may differentially impact women in correctional services [37] and should be considered when designing future research and mental health solutions. Consistent with previous research [2,38], the current results suggest that persons who were in married/common-law relationships were significantly less likely to screen positive for a mental health disorder than separated/divorced/widowed participants; contrasting previous research [2], the current results did not identify significant differences in positive screenings for people in a married/common-law relationship relative to people who were single. Factors specific to provincial correctional workers may interact with marital status and mental health and warrant additional research [38,39]. Positive screens for a mental disorder did not increase linearly as a function of age and years of service, contrasting results previous research [2]. The discrepant pattern may be due to occupational diversity among provincial correctional workers [30] or provincial correctional workers may reach a level of exposure that causes interacting risk and resiliency factors to differentiate people who stay in correctional work and people who leave. In any case, the discrepant pattern warrants additional research.

## 5. Limitations 

The current study had several limitations that provide directions for future research. First, the sample was self-selected and the sampling method prohibited knowing the actual response rate; as such, the reported proportions may not generalize to the provincial correctional population. Second, all participants were from Ontario; as such, the results may not generalize to other provinces or territories. Third, responses were based on anonymous online self-reporting of current symptoms, making reliability and validity ambiguous [40]; however, the results are consistent with previous Canadian correctional worker responses and increased reliability and validity would require a substantial investment. Fourth, even when anonymous, people may underreport clinical symptoms [41,42], and PSP, including correctional workers, report substantial concerns regarding stigma and confidentiality [5,6,43,44,45,46]. Fifth, even well-validated and conservative self-report screening tools are only approximations. Diagnostic interviews are a necessary next step justified by the current results. Sixth, focusing on current symptoms precluded lifetime assessments. Seventh, cross-sectional data prohibit discussions of causality with respect to potential risk and resiliency factors.

## 6. Conclusions

Many provincial correctional workers (58.2%) screened positive for one or more mental disorders, much higher than the diagnostic rates for the general population (i.e., ~10.1%) [3]. The frequency appears to be consistent with previous broad research involving correctional workers [2] and with results from CSC participants [4]. There were significant differences in positive screening frequencies between occupational categories among provincial workers that warrant further investigation. The current results are consistent with previous research, supporting the probability that the results are reliable and robust; in addition, the results emphasize the need for mental health supports among correctional workers. The results also further support calls for a national action plan emphasizing research, including a full epidemiology study, to support PSP mental health [47,48].

## Figures and Tables

**Table 1 ijerph-17-02203-t001:** Mean scores on mental disorder screening measures by occupational group.

Mental Disorder Screening Tools		Institutional					Probational Officers
	Total Sample	Wellness	Training, Chaplains, Coordinators	Governance	Correctional Officers	Administration	
	Mean (*SD*)	Mean (*SD*)	Mean (*SD*)	Mean (*SD*)	Mean (*SD*)	Mean (*SD*)	Mean (*SD*)
PTSD (PCL-5)	25.78 (20.38)	17.90 (15.71)	19.69 (19.18)	27.11 (17.89)	28.07 (21.18)	22.63 (21.34)	22.45 (19.33)
Depression (PHQ-9)	8.37 (6.56)	6.34 (5.51)	6.44 (6.63)	8.15 (5.87)	8.78 (6.79)	8.33 (6.74)	8.42 (6.34)
Anxiety (GAD-7)	7.01 (5.85)	5.67 (5.01)	4.94 (5.23)	6.72 (5.36)	7.34 (5.99)	6.13 (6.40)	7.29 (5.90)
Panic Disorder (PDSS-SR)	3.46 (5.20)	4.35 (5.68)	2.17 (4.41)	3.18 (4.48)	3.56 (5.27)	2.72 (5.66)	3.23 (5.17)
Alcohol Use Disorder (AUDIT)	5.77 (5.78)	3.36 (3.60)	3.45 (5.15)	6.57 (6.36)	6.48 (6.11)	3.04 (2.90)	4.93 (4.91)

*Abbreviations*. PTSD = Posttraumatic Stress Disorder; PCL-5 = Posttraumatic Stress Disorder Checklist for DSM-5 [10]; PHQ-9 = Patient Health Questionnaire [15]; GAD-7 = Generalized Anxiety Disorder Scale [28]; PDSS-SR = Panic Disorder Symptoms Severity Scale, Self-Report [19]; AUDIT = Alcohol Use Disorders Identification Test [25].

**Table 2 ijerph-17-02203-t002:** Total sample estimates by covariates and odds ratios between socio-demographic covariates and positive screenings for recent mental disorders.

Socio-Demographic Covariate	Any Positive Screen ^1^, % (*n*)	Odds Ratio (95% CI)
Sex		
Male (*n* = 448)	56.9 (255)	1.00
Female (*n* = 451)	59.4 (268)	1.11 (0.85, 1.45)
Age		
20–29 (*n* = 171)	53.8 (92)	1.00
30–39 (*n* = 266)	58.6 (156)	1.22 (0.83, 1.79)
40–49 (*n* = 239)	64.0 (153)	1.53 (1.02, 2.28) *
50–59 (*n* = 202)	56.4 (114)	1.11 (0.74, 1.68)
60 and older (*n* = 16)	31.3 (5)	0.39 (0.13, 1.17)
Marital status		
Married/Common-law (*n* = 553)	55.3 (306)	1.00
Single (*n* = 175)	57.7 (101)	1.10 (0.78, 1.55)
Separated/Divorced/Widowed (*n* = 127)	74.0 (94)	2.30 (1.50, 3.54) ***
Re-married (*n* = 33)	42.4 (14)	0.60 (0.29, 1.21)
Urban/Rural Work Location		
Urban (*n*= 874)	57.8 (505)	1.00
Rural (*n* = 24)	75.0 (18)	2.19 (0.86, 5.58)
Education		
High school or less (*n* = 42)	52.4 (22)	1.00
Some post-secondary (less than 4-year college/university program) (*n* = 422)	60.2 (254)	1.37 (0.73, 2.60)
University degree/4-year college or higher (*n*= 414)	57.0 (236)	1.21 (0.64, 2.28)
Years of service		
More than 15 years (*n* = 338)	62.1 (210)	1.00
10 to 15 years (*n* = 175)	64.6 (113)	1.11 (0.76, 1.62)
4 to 9 years (*n* = 114)	61.4 (70)	0.89 (0.63, 1.50)
Less than 4 years (*n* = 264)	47.3 (125)	0.55 (0.40, 0.76) ***
Occupational Group		
Institutional Wellness (*n* = 71)	43.7 (31)	1.00
Institutional Training, Chaplains, Coordinators (*n* = 26)	50.0 (13)	1.29 (0.52, 3.18)
Institutional Governance (*n* = 82)	61.0 (50)	2.02 (1.06, 3.85) *
Institutional Correctional Officers (*n* = 553)	59.0 (326)	1.85 (1.13, 3.05) *
Institutional Administration (*n* = 25)	52.0 (13)	1.40 (0.56, 3.49)
Probational Officers (*n* = 144)	63.2 (91)	2.22 (1.24, 3.95) **

^1^ Any positive screen and the total number of positive screenings include respondents who screened positive on any of the established mental disorder (i.e., PTSD, anxiety, panic disorder, alcohol abuse) screening tools and/or who self-reported being diagnosed with a mental disorder (i.e., obsessive-compulsive disorder, social anxiety disorder, persistent depressive disorder, Bipolar I, Bipolar II, cyclothymic disorder). * *p* < 0.05, ** *p* < 0.01, *** *p* < 0.001—Statistically significantly different from the reference group.

**Table 3 ijerph-17-02203-t003:** Frequencies of positive screenings for recent mental disorders based on self-report measures by occupation group.

Mental Disorder Screening Tools	Institutional						Probational Officers ^f^	
	Total Sample	Wellness ^a^	Training, Chaplains, Coordinators ^b^	Governance ^c^	Correctional Officers ^d^	Admin.^e^		Significant Differences Across Occupation Categories
	% (*n*)	% (*n*)	% (*n*)	% (*n*)	% (*n*)	% (*n*)	% (*n*)	
PTSD (PCL-5)	30.7 (302)	16.7 (14)	18.8 (6)	34.4 (31)	34.2 (202)	30.0 (9)	25.5 (40)	a < c, d; f < d
Major Depressive Disorder (PHQ-9)	37.0 (363)	21.3 (17)	28.6 (10)	35.5 (33)	39.7 (230)	40.0 (12)	37.4 (61)	a < c, d, e, f
Generalized Anxiety Disorder (GAD-7)	30.5 (292)	25.6 (20)	20.6 (7)	26.9 (25)	32.0 (180)	23.3 (7)	33.1 (53)	N/S differences
Panic Disorder (PDSS-SR)	14.1 (126)	21.7 (15)	--- ^5^	11.5 (10)	14.2 (77)	--- ^5^	12.4 (17)	N/S differences
Alcohol Use Disorder (AUDIT)	6.7 (61)	--- ^5^	--- ^5^	6.9 (6)	8.5 (46)	--- ^5^	5.2 (8)	N/A
Any other self-reported mood disorder ^1^	3.3 (30)	6.7 (5)	--- ^5^	--- ^5^	2.2 (12)	--- ^5^	7.2 (11)	N/A
Any positive screen for a mood disorder ^2^	39.4 (376)	25.6 (20)	29.4 (10)	38.2 (34)	41.5(235)	41.4 (12)	40.9 (65)	a < d, f
Any positive screen for an anxiety disorder ^3^	35.5 (313)	35.7 (25)	24.1 (7)	31.0 (27)	36.0 (191)	25.9 (7)	40.6 (56)	N/S differences
Any positive screen for any mental disorder ^4^	58.2 (524)	43.7 (31)	50.0 (13)	61.0 (50)	59.0 (326)	52.0 (13)	63.2 (91)	a < c, d, f
Total Number of Positive Screens ^4^								
0	46.7 (377)	61.5 (40)	54.2 (13)	41.6 (32)	45.7 (227)	50.0 (12)	44.2 (53)	
1	14.9 (120)	7.7 (5)	--- ^5^	18.2 (14)	15.3 (76)	--- ^5^	15.0 (18)	
2	11.5 (93)	12.3 (8)	--- ^5^	15.6 (12)	10.5 (52)	--- ^5^	13.3 (16)	
3 or more	26.9 (217)	18.5 (12)	20.8 (5)	24.7 (19)	28.6 (142)	25.0 (6)	27.5 (33)	

*Abbreviations*. Admin. = administration; PTSD = posttraumatic stress disorder; PCL-5 = Posttraumatic Stress Disorder Checklist for DSM-5 [10]; PHQ-9 = Patient Health Questionnaire [15]; GAD-7 = Generalized Anxiety Disorder Scale [28]; PDSS-SR = Panic Disorder Symptoms Severity Scale, Self-Report [19]; AUDIT = Alcohol Use Disorders Identification Test [25]. NS = Not significant; NA = Not assessed (due to low cell counts across occupation categories). ^a–f^ Lettered superscripts indicate categories of correctional workers that are significantly different from one another at *p* ≤ 0.05. ^1^ Any other self-reported mood disorder includes persistent depressive disorder, bipolar I, bipolar II, and cyclothymic disorder. ^2^ Any positive screen for a mood disorder includes all self-report mood disorders plus a positive depression screen (PHQ-9). ^3^ Any positive screen for an anxiety disorder includes positive screen for anxiety and panic disorder plus self-report obsessive-compulsive disorder and social anxiety disorder. ^4^ Any positive screen and the total number of positive screenings include respondents who screened positive on any of the established mental disorder (i.e., PTSD, anxiety, panic disorder, alcohol abuse) screening tools and/or who self-reported being diagnosed with a mental disorder (i.e., obsessive-compulsive disorder, social anxiety disorder, persistent depressive disorder, Bipolar I, Bipolar II, cyclothymic disorder). ^5^ Not presented due to insufficient sample size (i.e., *n* < 5).

**Table 4 ijerph-17-02203-t004:** Unadjusted odds ratios for positive screens of any current mental disorder ^1^ on sex by occupational group.

Sex	Total Sample	Institutional					Probational Officers
		Wellness	Training, Chaplains, Coordinators	Governance	Correctional Officers	Administration	
	OR (95% CI)	OR (95% CI)	OR (95% CI)	OR (95% CI)	OR (95% CI)	OR (95% CI)	OR (95% CI)
Male	1.00	1.00	1.00	1.00	1.00	1.00	1.00
Female	1.11 (0.85, 1.45)	1.30 (0.38, 4.45)	---^2^	0.54 (0.22, 1.33)	1.48 * (1.04, 2.12)	---^2^	0.87 (0.38, 1.99)

^1^ Any positive screen and the total number of positive screenings include respondents who screened positive on any of the established mental disorder (i.e., PTSD, anxiety, panic disorder, alcohol abuse) screening tools and/or who self-reported being diagnosed with a mental disorder (i.e., obsessive-compulsive disorder, social anxiety disorder, persistent depressive disorder, Bipolar I, Bipolar II, cyclothymic disorder). ^2^ Could not calculate due to low cell count size. * *p* < 0.05.

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
