# Peer review of "Provincial Correctional Service Workers: The Prevalence of Mental Disorders"

_ijerph, 2020, doi:10.3390/ijerph17072203_

Round 1

Reviewer 1 Report

Authors have conducted a research that adds an important contribution to the knownledge about the prevalence and correlates of mental health difficulties among correctional service employees. Certain considerations to be taken into account:

- It´s not accurate to say that 1487 correctional services employees participated in the study. Only 1032 completed a significant part of the survey. And the real number of of participants in the study (those included in the analyses-complete cases) seems to be around 900, but it isn´t given in the manuscript. Authors should be clearer about this important point.

- It´s somehow diffficult to follow the symptomatology measures descriptions, as these are mixed with each other. Moreover the description of each instrument is too brief. An independent complete description of each scale should be provided, along with sample ítems and validity and reliability data on each scale/subscale.

Author Response

Reviewer 1, Comment 1: Authors have conducted research that adds an important contribution to the knowledge about the prevalence and correlates of mental health difficulties among correctional service employees.

We thank the reviewer for their positive feedback and support.

Reviewer 1, Comment 2: It´s not accurate to say that 1487 correctional services employees participated in the study. Only 1032 completed a significant part of the survey. And the real number of participants in the study (those included in the analyses-complete cases) seems to be around 900, but it isn´t given in the manuscript. Authors should be clearer about this important point.

We appreciate the reviewer's feedback and have revised the abstract as follows: "Correctional service employees in Ontario, Canada (n=1487), began an online survey available from 2017-2018 designed to assess the prevalence and correlates of mental health challenges. Participants who provided enough data for the current study (n=1032) included provincial staff working in institutional wellness (e.g., nurses) (n=71), training (e.g., program officers) (n=26), governance (e.g., wardens) (n=82), correctional officers (n=553), and administration (e.g., record keeping) (n=25), and probation officers (n=144)."

We also refer the reviewer and readers to the following sentence in the Procedure and Data section, "A total of 1487 people began the survey, but only 1032 respondents completed at least some of the mental disorders section."

Reviewer 1, Comment 3: It´s somehow difficult to follow the symptomatology measures descriptions, as these are mixed with each other. Moreover the description of each instrument is too brief. An independent complete description of each scale should be provided, along with sample ítems and validity and reliability data on each scale/subscale.

We are happy to provide added details about each measure and congruent with the request from reviewer 2 for additional details about each mental disorder we have added an example item for each measure; however, we wanted to check with the editor before proceeding any further, as doing so would significantly increase the length and would contrast several previous papers (including in IJERPH; doi: 10.3390/ijerph17041234) where the current format was used because the measures are relatively common. Can the editor please advise?

Reviewer 2 Report

The research is very interesting and adequate, with good structure and statistics.

It is recommended:

The expansion of the introduction with explanations of the characteristics of mental disorders, since only mention is made of them.

The expansion of the results with greater clarification of them, lack explanatory development.

How do the findings lead to action for improvement?

Please complete the abbreviations for references in accordance with the policy of journal.

Author Response

Reviewer 2, Comment 1: The research is very interesting and adequate, with good structure and statistics.

We thank the reviewer for their positive feedback and support.

Reviewer 2, Comment 2: The expansion of the introduction with explanations of the characteristics of mental disorders, since only mention is made of them.
We have parenthetically added symptom descriptions for each of the mental disorders as follows:

"The results varied substantially (i.e., Posttraumatic Stress Disorder [PTSD; e.g., symptoms of intrusions, avoidance, negative mood, alterations in arousal and reactivity] 15.0% to 29.1%; Major Depressive Disorder [e.g., depressed mood or diminished interest in activities] 24.0% to 59.7%; and anxiety [e.g., excessive anxiety and worry] 12.2% to 37.9%), included the only Canadian study [2] which was based primarily on federal correctional worker (i.e., Correctional Services Canada; CSC) data rather than provincial workers, and 54.6% screened positive for one or more mental disorders [3]."

"Clinically significant mental disorder symptom severity was assessed using the following self-report screening measures: the PTSD Check List 5 (PCL-5; e.g., “Repeated, disturbing, and unwanted memories of the stressful experience?”) [10-14]; the 9-item Patient Health Questionnaire (PHQ-9; e.g., “Feeling down, depressed, or hopeless.”) [15-18]; the Panic Disorder Symptoms Severity scale, Self-Report (PDSS-SR; e.g., “How many panic and limited symptoms attacks did you have during the past week?”) [19-21]; the 7-item Generalized Anxiety Disorder scale (GAD-7; e.g., “Feeling nervous, anxious or on edge”) [18,22,23]; and the Alcohol Use Disorders Identification Test (AUDIT; e.g., “How often do you have a drink containing alcohol”) [24,25]."

Reviewer 2, Comment 3: The expansion of the results with greater clarification of them, lack explanatory development.

We have reordered the results, added additional specifiers to the text regarding which tables are being referenced, and added additional detail describing the results presented in the tables. Please see the revised document for details.

Reviewer 2, Comment 4: How do the findings lead to action for improvement?

We have added made the following revisions to the Conclusions section and we are happy to make additional revisions as needed: "The current results are consistent with previous research, supporting the probably the results are reliable and robust; in addition, the results emphasize the need for mental health supports among correctional workers. The results also further support calls for a National Action Plan emphasizing research, including a full epidemiology study, to support public safety personnel mental health [47,48]."

Reviewer 2, Comment 5: Please complete the abbreviations for references in accordance with the policy of journal.

We had downloaded the Endnote Style from the journal website and used that style for the references, and it appears in the most recent version that the references are corrected, but we are happy to help further, if necessary.